# Antimicrobials and Antibiotic-Resistant Bacteria: A Risk to the Environment and to Public Health

**Liliana Serwecińska**

European Regional Centre for Ecohydrology of the Polish Academy of Sciences, Tylna 3, 90-364 Lodz, Poland; l.serwecinska@erce.unesco.lodz.pl

**Abstract:** The release of antibiotics to the environment, and the consequences of the presence of persistent antimicrobial residues in ecosystems, have been the subject of numerous studies in all parts of the world. The overuse and misuse of antibiotics is a common global phenomenon, which substantially increases the levels of antibiotics in the environment and the rates of their spread. Today, it can be said with certainty that the mass production and use of antibiotics for purposes other than medical treatment has an impact on both the environment and human health. This review aims to track the pathways of the environmental distribution of antimicrobials and identify the biological effects of their subinhibitory concentration in different environmental compartments; it also assesses the associated public health risk and government policy interventions needed to ensure the effectiveness of existing antimicrobials. The recent surge in interest in this issue has been driven by the dramatic increase in the number of infections caused by drug-resistant bacteria worldwide. Our study is in line with the global One Health approach.

**Keywords:** antibiotics; antimicrobial resistance; environment; contamination; risk assessment; one health

## 1. Introduction

Antibiotics, antimicrobial substances that have the capacity to inhibit the growth of microorganisms or kill them, are widely used for the treatment of bacterial infections in humans and animals, as well as in nonmedical applications. The global annual production of antibiotics is estimated to be as high as 100–200 thousand tons, with more than one billion tons having been produced since 1940 [1,2]. Such overuse has resulted in a substantial increase in the rates of antibiotic excretion and environmental release and hence the growth of drug resistance in bacterial strains; this represents a global problem and a considerable threat to human health. Almost 100 years have passed since Fleming's discovery of penicillin as a life-saving medicine in 1928, and humanity is again confronted with the lack of an effective weapon for fighting infections, despite an intensive search for new effective antibacterial drugs.

The last decade has seen increasing interest in the spread of bacterial resistance in the natural environment, and this interest has stemmed from growing concern among the medical and scientific community related to the rapid escalation of antibiotic-resistant bacteria (ARB), including resistance to a new generation of antibiotics and pharmaceuticals of last resort. Resistant bacteria are responsible for infections that are more difficult to treat, requiring the use of drugs that are more toxic and more expensive. In some cases, bacteria have become resistant to all known antibiotics [3]. The phenomenon of drug resistance is exacerbated by the fact that a wide variety of antibiotics are not only used for medical and veterinary purposes but also to promote the growth of livestock [4]. The massive scale of antibiotic use and antibiotic misuse accelerates the evolution of ARB (antibiotic resistant bacteria) and ARG (antibiotic resistance genes) in the environment, thus increasing the risk of transmission of the environmental resistome to humans [5]. The resulting growth in pathogen infections and

the problem of antibiotic-resistant bacteria have become global public health threats. This threat, moreover, has higher mortality and morbidity rates than those of HIV and prostate and breast cancers combined [6–8].

For many years, research has focused on the clinical aspect of the spread of bacterial antibiotic resistance; however, in the last decade, a growing number of environmental studies have examined the quantitative and qualitative aspects of antibiotic release into the environment. Antimicrobials and their bioactive metabolites are able to enter various environmental compartments via a range of pathways; they can reach water bodies through urban wastewater and agro-ecosystems through fertilization with antibiotic-polluted sewage sludge, manure, sediment, or biosolids. Thanks to their high water solubility, these ecotoxic compounds rapidly spread to aquatic and soil ecosystems, and their subsequent accumulation poses a great risk to the quality of the receiving waters, including groundwater and soil ecosystems. Next to antibiotic residues, ARB and ARG have also been recognized as emerging environmental pollutants that can spread rapidly across the globe.

Close international collaboration is, therefore, needed to address this problem [9]. A comprehensive approach would be desirable—more specifically, one that takes into account the lack of new antibiotics with a novel model of action, the excessive and irresponsible use of existing antibiotics, the distribution of antibiotics into the environment, e.g., due to failure of wastewater management, and their use in the animal and food industries. This approach could yield long-lasting benefits if taken soon. However, if left unchecked, it has been estimated that antimicrobial resistance could lead to 10 million deaths per year by 2050 [10,11].

## 2. The Fate of Antibiotics in the Environment and their Biological Effects

Three different routes for releasing antibiotics into the environment can be distinguished: feed additives for stockbreeding and fish aquaculture, human and veterinary drugs, and environmental release during production (Figure 1). Typically, 10–90% of the antimicrobials ingested by humans and animals are metabolized, and the remainder are excreted in the feces in an unchanged, i.e., still active, form, which can contaminate urban wastewater, manure, and biosolids. The rate of antibiotic excretion is believed to vary according to the chemical structure and applied dosage, as well as the animal age and species [12,13]. Unused or expired drugs are also often flushed into the sewage system, which places an additional burden on the wastewater system and, thus, on the environment. A significant amount of antibiotics, as well as their degradation products and bioactive metabolites, have been introduced into water and agro-ecosystems through fertilization and irrigation with antimicrobial-polluted sewage sludge, manure, biosolids, sediment, and reclaimed water, resulting in their accumulation in the water, including groundwater and agro-ecosystems [14–16].

Some natural environments, such as the areas around pharmaceutical plants and animal farms and aquacultures, as well as those receiving treated hospital wastewater, are constantly exposed to contaminants containing antibiotics, resulting in the persistence of varying concentrations of antimicrobials. Their biological activity in the environment depends on their bioavailable fraction and the local environmental conditions, including pH, water content, organic carbon content, and characteristics of the local microbial community. The impact of antibiotic residues on aquatic and terrestrial ecosystems is still not fully understood [17].

Regarding their persistence, antibiotics remaining in the natural environment may become a source of nutrients for some groups of microorganisms and become degraded; however, such biodegradation depends on many factors, such as the composition of the microflora, pH, temperature, and humidity (microbial biodegradation). Non-biotic degradation can also take place via hydrolysis, oxidation, reduction, or photolysis, and this also depends on many physicochemical properties and environmental conditions. Residues of synthetic and semi-synthetic antibiotics, such as fluoroquinolones and sulfonamides, are more chemically stable and less susceptible to bacterial degradation processes; thus, their residues are often detected in the environment [18–21]. A cluster of 90 pharmaceutical factories in India releases wastewater with a content of about 30 mg/L of ciprofloxacin, resulting in the deposition of

several kilograms into the environment each day and hence many tons a year; consequently, the waters of the lake near the factories have been found to demonstrate a ciprofloxacin concentration as high as 6.5 mg/L [22].

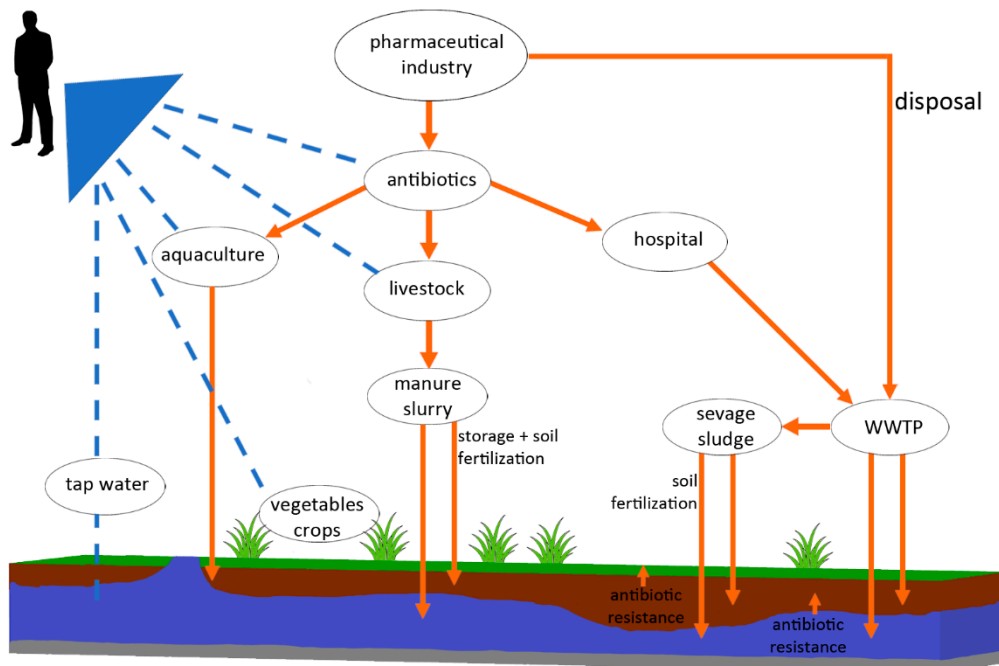

**Figure 1.** Release of antibiotic residues into the environment and human exposure to antimicrobials and antibiotic-resistant bacteria.

Once in the environment, antimicrobial residues have multiple deleterious impacts on biota at different trophic levels. When present at subinhibitory concentrations, antibiotics can function as signaling molecules and cause shift in bacterial gene expression and gene transfer, virulence, biofilm formation and quorum sensing, and the modulation of gene activity. In addition, through the prolonged exposure of bacterial cells to pharmaceutical substances, these substances can also accelerate the horizontal transfer of drug resistance genes (HGT-horizontal gene transfer) between microbes, often between phylogenetically-distant bacteria and between non-pathogenic bacteria and pathogens. These antimicrobials residues can exert an ecotoxic effect on non-target organisms living in both aquatic and terrestrial environments [23,24]. They can be absorbed by plants and interfere with their physiological processes, such as photosynthesis, germination, and growth [25,26]. Finally, these residues can exert an influence on human health through the consumption of contaminated water and agricultural products, resulting in the development of cancer, allergic reactions, or disorders in the composition of the natural intestinal microflora [27].

A balance exists between the bacterial strains inhabiting the human gastrointestinal tract—95% of which are beneficial, and the remainder of which are opportunistic pathogens—as well as between these bacteria and the human body [28,29]. Many clinical and epidemiological studies indicate that antibiotic consumption can alter the composition of the intestinal microbiota, resulting in the emergence of antibiotic resistant-bacteria and the proliferation of opportunistic pathogens. This alteration can also be responsible for conditions such as intestinal disorders, pseudomembranous colitis, and colorectal cancer [30]. Furthermore, such imbalances in the gut flora can dysregulate development of the immune system, which may affect adiposity and bone growth; this is particularly unfavorable in the early years of life [31,32]. Thus, the presence of subinhibitory concentrations of antibiotics often found in contaminated areas has serious implications for the sustainability of the environment and human health.

### 3. Sub-Inhibitory Concentrations of Antibiotics in the Environment and their Biological Effects

The Minimal Inhibitory Concentration (MIC) has been defined as the lowest antibiotic concentration that prevents visible microorganism growth after incubation with bacteriological media and falls within the range of clinical dosages of antimicrobial drugs. In sub-inhibitory concentrations (concentrations below the MIC), the susceptible bacterial cells continue growing at a reduced growth rate, and the lowest antibiotic level that drives selection of a resistant mutant over the wild type cells is determined as the Minimal Selective Concentration (MSC). It is estimated that the MSC for a diverse range of microorganisms ranges between 1/4 and 1/230 of the MIC values [33,34]. Moreover, it has been documented that resistance that evolves as a consequence of sub-clinical levels is likely to have a lower cost on bacterial fitness compared to resistance as a result of exposure to clinical levels of antibiotics during therapy [35].

A number of studies have examined the results of the long-term exposure of environmental bacterial strains to low concentrations of antibiotics. Such exposure appears to have a significant influence on bacterial genomes. For example, this exposure has been found to modulate the transcription levels of about 5–10% of bacterial genes. These genes are associated with a variety of cellular processes, such as protein synthesis and carbohydrate metabolism—not just processes associated with the target of antibiotic action [36–39]. Sub-inhibitory concentrations of antimicrobials can induce the bacterial SOS repair system, which, in turn, increases the frequency of genome mutation and horizontal gene transfer and mobile genetic elements, including those responsible for antimicrobial resistance [40,41]. *P. aeruginosa* strains, known to be abundant in wastewater, produce more intense biofilm after exposure to sub-inhibitory doses of erythromycin, sulfamethoxazole, or roxithromycin [42]. *Staphylococcus aureus* strains have also demonstrated the overexpression of genes encoding certain factors that may increase virulence; such changes have serious clinical consequences, as they may increase the risk of illness and even mortality among patients [43,44]. The results of a survey of sub-inhibitory concentrations of antibiotics in environmental samples was reported by Chow et al. In this study, the authors collated 40 scientific papers, covering the period from 1999 to 2018, that reported measurements of antibiotic levels in diverse environments; it was found that the environmental concentrations of antibiotics in many samples fall into the MSC range and are likely to be influencing bacterial ecology and triggering the selection of antibiotic resistant bacterial cells. Data were collected for as many as 39 different antibiotics belonging to 9 different classes. Drug concentrations ranged from $10^6$ ng/L to $10^{-2}$ ng/L. In a significant number of samples, the concentration values exceeded the MSC values, and approximately 2% of them overlapped even the MICs noted for a diverse range of microorganisms. In the same study, the authors also highlighted the need for accessible and accurate testing methods to determine low concentrations of antibiotics in environmental samples, particularly for in-field measurement equipment [45]. Some researchers emphasized that the minimum concentration of some antimicrobials required for detection is higher than the concentration that causes a biological effect (1/4–1/230 the MIC ). These antibiotics could be present in the environment at undetectable (by commonly used techniques), but still relevant, concentrations [33–46]. For example, the MIC of ciprofloxacin for 82 tested bacterial species ranged between 0.002 and 4 mg/L [47]. However, the limit of detection with HPLC (High-Performance Liquid Chromatography) was 0.005 mg/L [15].

It should be noted that the *de novo* emergence of bacterial resistance can develop from single bacterium and be disseminated worldwide. One such documented example is the mcr-1 gene, which confers resistance to colistin and spread through a single *de novo* mutation event that likely occurred in China and was triggered by the extensive use of colistin in swine farms [48–50]

### 4. Antibiotic Resistant Bacteria (ARB)

The changes mentioned in the previous section have been attributed to the induction of various types of responses to environmental stresses in microbial cells. All these changes serve as an expression of adaptation to new, unfavorable conditions and the emergence of new bacterial phenotypes [51–53]. Bacterial genomes may feature mutations or genes that prove to be advantageous for their survival in

the presence of antimicrobial agents. Antibiotic-sensitive bacterial cells, in turn, may become resistant via *de novo* gene mutation or through the acquisition of resistance genes from other bacterial cells. Hence, massive overuse of antibiotics results in the acquisition of resistance through selection pressure [54]. As mentioned above, antibiotic resistance can be acquired via horizontal gene transfer (HGT) between cells, even between those of different species or genera [55,56]. HGT is believed to take place via mobile genetic elements, such as plasmids, transposons, integrons and prophages, as indicated by the fact that identical sequences of drug-resistance genes have been identified in the DNA of environmental and clinical bacterial strains [17,57]. Currently, the introduction of a new antibiotic into the market is almost immediately followed by the emergence of resistant bacterial strains. Hence, a full understanding of the mechanisms and spread of drug resistance is crucial for the development of new effective antibacterial substances and antibiotics [4]. It should be emphasized that resistant bacteria are naturally present in the environment, and ARGs have been identified in ancient DNA recovered from both human ancestor and environmental samples, suggesting that bacterial antibiotic resistance mechanisms predate the use of antimicrobials by humans [58–63]. However, studies have also identified higher numbers of ARGs in the genomes of modern strains of some bacteria (e.g., the genera *Pseudomonas* and *Clostridium*) than in strains recovered from the microbiome of ancient human ancestors, suggesting that resistance gene transfer and large-scale antimicrobial use have increased the acquisition of ARGs [63].

One area of particular concern is the rapid spread of multi-drug resistance (MDR) in numerous countries, some cases of which have no available treatment. While MDR is defined as the acquired non-susceptibility to at least one agent among three or more antimicrobial classes, extensive drug resistance (XDR) indicates non-susceptibility to at least one agent in all but one or two antimicrobial classes and pan-drug resistance (PDR) as non-susceptibility to all agents in all available antimicrobial classes [64]. Natural reservoirs and habitats of multi-drug resistant bacteria (MDRB) include water, soil, wastewater, sewage, plants (fruit, vegetables, herbs), raw meat, and dairy products, as well as the gastrointestinal tract, the upper respiratory system, and the skin of humans and animals. Livestock is also a well-known reservoir for MDRB, especially pigs, cattle, and poultry. The transmission of MDRB via water and food products is also possible. Numerous publications have demonstrated that drinking water, milk, and meat products can be contaminated with MDRB [48,65,66]. Knowledge about the pathways and transmission mechanisms of bacterial drug resistance genes in various niches and bacterial populations, including clinical and environmental ones, is still being updated. For example, studies conducted in Romania identified resistant *Salmonella* bacteria in about 13% of raw chicken meat. These bacteria constitute a reservoir of genes determining resistance to antibiotics, including tetracycline (66.6% of isolates), nalidixic acid (64.3%) sulfamethoxazole (64.3%), ciprofloxacine (61.9%), strteptomycin (59.5%), trimethoprim (33.3%), ampicillin (9.5%), chloramphenicol (7.1%), and gentamicin (2.4%) [67].

Some resistant bacteria are naturally present in the environment and also have clinical importance. The European Antimicrobial Resistance Surveillance Network, a group that reports resistance trends for Gram-negative bacteria, has highlighted the urgent need to slow the phenomenon of drug resistance in the natural environment. Gram-negative bacteria are naturally resistant to vancomycin due to their cell wall properties. In addition, *Klebsiella* spp. exhibit insensitivity towards ampicillin, and *Pseudomonas aeruginosa* strains are usually resistant to tetracycline, chloramphenicol, sulphonamides, and trimethoprim [68].

Among Gram-negative bacteria, the genera *Pseudomonas* and *Acinetobacter* and those of the *Enterobacteriaceae* family have been prioritized by the World Health Organization (WHO) as taxa for which there is an urgent need to develop new, effective drugs. These groups of Gram-negative bacteria are of particularly high concern as agents of antimicrobial resistance for four main reasons. Firstly, they produce an extended spectrum of β-lactamases (ESBLs) that confer resistance to such antimicrobials as cephalosporins, penicillins, and monobactams, and they include a growing number of carbapenem-resistant strains; all of these are new-generation antibiotics that are employed as "the last line of antibiotic defense" against resistant organisms [69,70]. Worryingly, a dramatic increase in

nosocomial CRB (carbapenem-resistant bacteria) infections has been recorded worldwide over the last decade, with infections by *Acinetobacter* and *Pseudomonas* demonstrating 40–80% mortality in intensive care units [71–73]. Secondly, many of these species are opportunistic pathogens that infect the sick or those with weaker immune systems. Thirdly, they are also associated with nosocomial infections, sepsis, wound infections, urinary tract infections, and secondary pneumonia. Finally, many of these bacteria appear to be widely distributed in nature, and their growth may be enhanced in a contaminated environment.

The Gram-positive methicillin resistant *Staphylococcus* aureus (MRSA) and vancomycin-resistant Enterococcus (VRE) bacteria are also of great concern for public health. Both are known to have the potential to cause epidemics [74–76]. Although few studies have confirmed the presence of ESBL, MRSA, and VRE producers in the environment, where they can act as a reservoir of such resistance, their presence is nonetheless disturbing, as these genes can be transferred into human pathogens [77–80]. A number of MDR bacteria have been isolated from hospital and municipal sewage, as well as from the soil around animal farms and polluted rivers, suggesting that they may play a role in the dissemination of antibiotic resistance and may become potential pathogens. Their growing clinical importance, associated with their resistance to various antibiotics, has resulted in them adopting the characteristics of both environmental and clinical bacteria [81–84].

## 5. Antibiotics, ARB, and ARG in Water, Wastewater, and Sewage Sludge

Wastewater resulting from human activities, such as wastewater from industrial plants, healthcare services, agriculture, and the general population, is collected at wastewater treatment plants, turning such plants into point sources for antimicrobials, antibiotic-resistant bacteria (ARB), and antibiotic resistance genes (ARG) [85]. The main sources of antimicrobials in surface waters and groundwater are effluents from wastewater treatment plants (WWTPs), pharmaceutical facilities, and surface run-off from animal feeding operations [57,86]. Although some biological and physicochemical methods, such as biodegradation, adsorption on activated carbon, membrane separation, and chlorine disinfection are commonly used to remove antibiotics in WWTPs, numerous studies have found these measures to only be partially effective. However, in some cases, these methods may not be employed due to the high costs associated with their use, and the degree of degradation of an antimicrobial varies according to the group of drugs and their chemical structures [86–89]. In addition, many conventional WWTPs are not designed for the removal of pharmaceuticals, and antibiotic residues have been detected in the final effluents of WWTPs around the world. An important issue and emerging problem are hospital wastewaters that contain pathogens and a wide range concentrations of antibiotics, which do not have a regulatory status (in most countries, hospital effluents are not classified as separate effluents with a particular hazard to the environment, and they enter municipal WWTPs. Therefore, a monitoring system is necessary for these effluents [90,91].

Due to the growth in the quantities of produced wastewater and its associated sewage sludge, the issue of proper sludge management has become an important element of contemporary environmental research [92]. Due to its high abundance of organic matter and nutrients, sewage sludge is commonly disposed of by spreading it over soil as part of agricultural management; however, sewage sludge is characterized by high levels of ARBs and ARGs removed from the incoming wastewater, which can pose a significant environmental and human health hazard [93,94]. The selective pressure that antibiotic pollution may exert on antimicrobial proliferation in soil is of particular concern, as soil contains a large number of bacterial genera of clinical relevance [95,96]. A study conducted in The Netherlands in 2008 found soil samples to contain up to 15 times more ARGs than soil samples from 1970 [97]. The xenobiotics and resistant microorganisms present in sludge can shape the metabolic activity and diversity of the soil microbial community, thus negatively affecting the quality of the soil ecosystem [97].

This problem is compounded by the fact that some antibiotics, such as fluoroquinolones, tetracyclines, and sulfonamides, bind to soil particles, which significantly slows and hinders the

process of their biodegradation. These antibiotics remain in the soil as stable non-degraded molecules and can easily pass into the surface water and groundwater. Furthermore, synthetic and semi-synthetic antibiotics, e.g., fluoroquinolones, are less susceptible to bacterial processes: For example, ciprofloxacin and oxolinic acid require five months for only 20% of oxolinic acid to be removed [22]. Recent reports also highlight the need to expand the list of emerging contaminants present in sediments with ARB and ARG and to eliminate them from both sediments and fertilized soils [2,98].

Antibiotics have frequently been detected in surface water at concentrations generally ranging between 0.01 and 1.0 μg/L. This suggests that they have entered the sources of drinking water and may not be completely removed by traditional drinking water treatments. In addition, they may persist in the water for months. The most frequently detected antibiotics are quinolones, macrolides, sulfonamides, and chloramphenicols in drinking water, with ciprofloxacin dominating in the quinolones group [1]. In 2015, sulfapyridine, sulfamethoxazole, ciprofloxacin, enrofloxacin, levofloxacin, norfloxacin, chloramphenicol, florfenicol, doxycycline, and metronidazole were detected in tap water at concentrations ranging between 0.5 and 21.4 ng/L in rural Shandong province, eastern China [99]. Water from a river system in Germany supplying a drinking water reservoir was found to contain residues of sulfamethoxazole (up to 0.40 μg/L), trimethoprim (up to 0.39 μg/L), and macrolide antibiotics, such as clarithromycin (up to 0.60 μg/L), downstream from two sewage treatment plants [100]. A study of two treatment plants in Bagdad, Iraq, identified ciprofloxacin, levofloxacin, and amoxicillin in raw water and ciprofloxacin and levofloxacin in the finished water [101]. In addition to the fact that these drugs emit unpleasant odors and can cause skin disorders, long-term exposure to ciprofloxacin and other antibiotics in potable water sources represents a serious health problem, even at low levels [102]. Some studies have analyzed changes in the total (i.e., intrinsic and acquired) resistance of autochthonous microorganisms in rivers that act as receivers of treated wastewater. One study found the abundance of tetracycline-resistant and fluoroquinolone-resistant bacteria to increase in river water samples further downstream [103]. Multi-drug resistant opportunistic bacteria carrying genes resistant to carbapenem, a last-resort antibiotic, have also been found in surface waters such as ponds, lakes, and rivers, representing a major challenge to public health [84,104,105]. The concentrations of ARBs and ARGs are usually higher at the point of effluent discharges from WWTWP but gradually become reduced downstream from the discharge point [106,107]. A decline in the concentration of ARGs and ARBs downstream of the WWTW effluent discharge can occur as a result of different factors, such as transport, dilution, degradation, and adsorption [108,109].

## 6. Antibiotic Residues and Antimicrobial Resistance in Agriculture

The rapid growth of stockbreeding has raised several concerns regarding the quality and safety of animals. As such intensive production entails higher concentrations of animals in small spaces under crowded and stressful conditions, which substantially increases the risk of infectious diseases, antibiotics have been used intensively in the sector since the 1950s [110]. Initially, these antibiotics were prescribed by veterinarians to treat animal diseases such as chronic enteritis, which cause poor nutrient absorption and reduced weight gain in animals. However, antibiotics soon began to be used on a massive scale at sub-therapeutic doses for prophylactic and metaphylactic purposes as ASGs (antibiotic stimulators of growth) to increase the growth of farm animals. Presently, antibiotics are commonly used as a prophylaxis, i.e., to prevent the development of infections, and metaphylaxis, i.e., for treating sick animals. Both approaches are intended to avoid the rapid spread of contagious diseases, especially in countries where no other preventive methods are employed. Among these antibiotics, penicillin was the first to be used on a large scale in the nutritional practice of animals, followed by more potent compounds, such as oxytetracycline, chlortetracycline, flavophospholipid, and bacitracin [110]. The most commonly used feed antibiotics in the European Union are salinomycin, avoparcin, monosin, tylosin, spiramycin, and virginiamycin. Although it is difficult to estimate the true amounts of antimicrobial substances used in food-producing animals due to the differences in distribution and registration systems between countries, the data suggest that about 80% of all antibiotics sold in 2009

in the USA were used for animal production [111]; in the EU, nearly nine tons were used for animal production in 2014, while less than four tons were used in human medicine [110]. This is of significant importance because almost all of the active antimicrobials used in animal husbandry are structurally related to those used for treating people, which promotes co-resistance and cross-resistance [110,112]. In addition, six classes of antibiotics commonly used in agriculture and aquaculture (penicillins, tetracyclines, aminoglycosides, macrolides, quinolones, and sulphonamides) have been reported by the WHO to be extremely important antimicrobials for human treatment [113], and 37 of the 51 antibiotics used by the main animal and aquaculture producing countries are indicated as highly important or critically important in human medicine [114]. Recent microbiological and clinical evidence suggests that ARBs and ARGs are often transferred from food-producing animals to humans [115]. In addition to breeding cattle, pigs, poultry, and rabbits, some antibiotics are also used for preventive purposes in plant and fruit crops, as well as bee breeding [116,117]. There are numerous studies on the major factors in the selection and dissemination of food borne antimicrobial resistance along the food chain. The major area of investigation is the livestock from farms through slaughterhouses, processing plants, food packing points, and retail sales. Pathogenic and non-pathogenic bacteria resistant to antibiotics can be transmitted from livestock to humans via food consumption or via direct contact with animals [118]. Any food contaminated with resistant bacteria provides a direct route for human colonization [119–121]. Some authors concluded that the transmission of ARB via food is likely to be the most important pathway from livestock [122]. Pathogenic bacterial strains resistant to one or more antibiotics are also isolated from fish, poultry, milk, dairy products, and vegetables. Some types of resistant bacteria that cause serious infections in humans have developed resistance to most of the available antimicrobials [123–127].

## 7. Antimicrobials in Aquaculture

The most rapidly-growing animal food-producing sector is aquaculture (www.fao.org). Its contribution to the total amount of fish intended for human consumption rose from 5% in 1962 to 44.1% in 2014 [128]. It has been estimated that global fish production amounts to about 110 million metric tons per year; of this supply, approximately 50% is derived from aquaculture production [129]. The rapid transition from a fishing to a production model occurred as a response to the depletion of fish stocks through overfishing and global climate change. However, antibiotics are also used on a wide scale in intensive aquaculture, mainly fish, shrimp, and shellfish, resulting in extremely large amounts of antibiotics entering the aquatic environment [130]. Drugs applied to fish feed also can persist in the aquatic ecosystem for a long time. In addition, fish do not metabolize antimicrobials effectively: It has been determined that about 70–80% of active antibiotics and their metabolites are suspended in water and can spread throughout the water system Therefore, the practice of using antibiotics in fish production has serious environmental consequences, as it may result in selection pressure in many ecosystems [131,132]. As such, sediments and watercourses demonstrate high proportions of antibiotic-resistant bacteria and can serve as sources of antibiotic resistance genes for fish pathogens. Furthermore, it has been found that resistant bacteria from aquaculture may be transferred to terrestrial animals and the human environment and are capable of transferring their resistance genes to opportunistic animal or human pathogens. The passage of ARBs and ARGs from aquatic ecosystems has been found to affect the health of humans and terrestrial animals. None of the antibiotics used in aquaculture, mainly for prophylactic and metaphylactic purposes, are specifically dedicated to this purpose. Instead, they are drugs developed for other areas of veterinary medicine [130]. As such, the same resistance patterns observed in terrestrial animal farming are also found in aquaculture [133–135]. Furthermore, the antimicrobials applied in aquaculture are typically applied as a metaphylaxis to entire populations containing healthy, sick, and carrier individuals and at doses that are usually higher than those used in terrestrial animal farming. Florfenicol, oxytetracycline, sulfadimethoxine, and ormetoprim are authorized for use in aquaculture in the USA [131], while florfenicol oxytetracycline, erythromycin, sarafloxacin, sulphonamides, trimethoprim,

and ormetoprim are authorized in most European countries [17]. China demonstrated the highest production and use of antibiotics until 2010 [136], at which time, the country also accounted for 67% of the total worldwide aquaculture and was the largest exporter of fish and fishery products [128]. However, China does not require veterinary prescriptions for the use of antibiotics in animals [137]. China is followed by India (where antibiotics sales and usage are also not regulated), accounting for 8% of total worldwide aquaculture production [138].

## 8. Government Policy Interventions to Protect Antimicrobials

Despite scientific and political awareness, as well as increasing attention by the mass media, the prevalence of bacterial antibiotic resistance continues to grow, and a global effort is required to control it. The spread and maintenance of clinically-relevant ARBs in the environment, driven by the ongoing release of antimicrobials, is closely correlated with antibiotic consumption patterns. Recent analyses suggest that human antimicrobial consumption increased by 65% between 2000 and 2015, with a rapid increase of last-resort antibiotics such as carbapenems, polymyxins, oxazolidinones, and glycylcyclines. Approximately 35,000 people in the U.S and 33,000 in the EU die every year due to infections by antibiotic-resistant bacteria. Furthermore, in 39% of these cases, death was associated with infections by bacteria resistant to last-resort antibiotics [138,139]. The importance of this issue was recognized by the international community in September 2015, when the United Nations (UN) Member States adopted the 2030 Agenda for Sustainable Development [140]. Recent predictions suggest that AMR could impede Sustainable Development Goals due to its serious consequences for human health, social well-being, and economic development. As such, an effective response based on multi-sectoral cooperation is needed [140–143]. So far, various types of government policies have recently been established to reduce the overuse and misuse of antimicrobials. Since 2006, in all Member States of the European Union, the use of any growth promoters has been prohibited. In addition, political initiatives concerning AMR have been developed, such as the 2016 United Nations resolution [20,144], the 2017 Berlin Declaration of the G20 Health Ministers [145], the recent draft opinion on ABR by the 2018 European Parliament (which emphasized that the routine collection and reporting of monitoring data at the EU level should be mandatory [146], and the 2008 "European Antibiotic Awareness Day", an initiative that provides a platform for national campaigns to raise awareness about the prudent use of antibiotics. All the above are promising signs that governments and the international community are mobilizing to act on AMR. However, despite the use of strategies like antimicrobial guidelines, public awareness campaigns, and changing the prescribing behaviors of individual physicians, most countries have not yet started implementing government policies to create large-scale reductions in antimicrobial use through population-wide interventions [76,147]. Attempts have been made to create a systematic review and evidence map for identifying, describing, and assessing the full range of implemented government policy interventions aimed at reducing antimicrobial use. Seven databases were searched up to January 28 2019. The resulting studies were organized according to the following policy categories: regulatory interventions, guidelines, communication policies, legislation, environmental/social planning interventions, and fiscal interventions. The authors identified 69 evaluation studies examining the impact of policy interventions on antimicrobial usage across four of the WHO's six regions that reported 17 different policy strategies. The findings indicate that most existing policy options have not been rigorously evaluated, and some commonly-discussed policy options have not been evaluated for their impact on antimicrobial use. To avoid wasting public resources, governments should ensure that future AMR interventions are evaluated using rigorous study designs and that the relevant study results are published [148]. Legislation is needed to clarify the responsibilities of governments in the fight against antibiotic resistance. In 20 countries throughout the WHO European Region, a high level of specific regulations was found to significantly correlate with lower antibiotic consumption [149]. If appropriate steps are taken soon, the costs could be inconsiderable, while the benefits would be long lasting.

## 9. Conclusions

Recent years have seen a considerable rise in the number of pathogens with multi-drug resistance genes, believed to be associated with the mass production of antimicrobial substances, their excessive use for medical purposes, particularly in food-producing animals, combined with the many pathways of their release into the environment. Such resistance now represents a serious public health threat. To counter this trend and protect the effectiveness of existing antibiotics, current efforts are focused on limiting the large-scale and inappropriate use of antibiotics. The low antibiotic concentrations in the environment have significant biological effects and can still promote the acquisition of resistance via selection pressure. However, detailed knowledge about the rates of dissemination and degradation of different antibiotic classes in the environment must be acquired. The key questions are as follows: What are the concentrations of various antibiotics in environmental samples, and what levels are needed to stimulate the acquisition of resistance by microbial cells?

Notably, there is an ongoing search for new antimicrobial drugs to combat infections caused by resistant pathogens. Research groups around the globe are suggesting alternative solutions for treating resistant organisms, and a post-antibiotic golden age characterized by the use of biological, biogenic, or bio-based products and therapies is eagerly awaited. However, regardless of such efforts, there is a pressing need to support public health and retain the effectiveness of currently-used antibiotics. To this end, it is vital to keep the environment free of antibiotic residues and their active metabolites, which are known to be responsible for the accumulation of drug resistance. Therefore, efforts should be directed towards acquiring knowledge of antibiotic use trends in different regions of the world and the emergence of drug resistance at the clinical and environmental levels. Furthermore, interventions and routine monitoring programs should be prioritized, together with public education campaigns at all levels, such as healthcare, community, and individuals.

**Funding:** This research received no external funding.

**Conflicts of Interest:** The author declares no conflict of interest.

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
