# Peer review of "Antimicrobials and Antibiotic-Resistant Bacteria: A Risk to the Environment and to Public Health"

_water, doi:10.3390/w12123313_

Round 1

Reviewer 1 Report

The review deals with a topic of global relevance and interest. It is well written and easy to follow as it has a clear thread. Also, it is scientifically sound.
However, for a review of such a relevant and recent topic, it seems to me that some of the literature is a bit outdated, and that more recent work about the presence of antibiotics and resistance genes in the environment should be incorporated. Most of the cited works ranged from 2011-2015, there are several recent works 2018-2019, concerning this topic that can be included in order to enhance the novelty of the review. 
On the other hand, the conclusion should be more robust, incorporating the different aspects or topics included in the development of the review.

Reviewer 2 Report

This is an interesting review that focus on the potential spread of AMR through the environment, especially waters.

Though not original, the topic always deserves attention and awareness, and therefore such kind of reviews are welcome.

The major concern of this paper stands on its overall structure: the first chapters are quite redundant and repetitive; the concepts are absolutely important, but are melted into and with others; to give an example, the chapter entitled "sub-inhibitory concentrations..." starts dealing with such a topic at line 101, more than half-length of the chapter; the difficult degradation of ciprofloxacin (not ciproflaxacine) in the soil is repeated at least twice, if not more, and other considerations on the diffusion of antimicrobials or ARG within the environment do the same.

Abbreviations and references style should carefully be revised, as they do not match the requirements.

I would suggest the author to rewrite this interesting review in a completely more straightforward way, and to avoid redundancies in order to catch the readers' attention and make the reading more attractive.

Reviewer 3 Report

Comments

Lines 41-42: Referring to antimicrobials in personal care products (which are surely antiseptics) in the middle of references to antibiotics is misleading. It implies that antibiotics are in these products. I think the author should avoid blurring the lines between antibiotics and non-antibiotic antimicrobials.

Lines 46-47: awkward. I suggest "Infections due to ARB result in higher mortality and morbidity rates than HIV..."

Line 62: "relatively inexpensive". This statement is not well supported. In the rest of the review, the author discusses the various inputs of antibiotics into the environment including manufacturing, human overuse, animal husbandry, and aquaculture. The required pollution controls and possible reduction in animal size and fish survival could well be costly. Additionally, expense is not the only problem, as coordination and cooperation among governments and between governments and industry could be difficult as the authors discuss near the end. I would suggest that the author modify this statement or provide some evidence that this would be an inexpensive fix.

Line 83: "constant persistence" is redundant.

Line 103: "been found to cause alterations.."

Line 214: "as well as"; not "week'"

Line 246: This paragraph discusses concentrations of antibiotics found in different environmental locations. An earlier paragraph discussed effects of subinhibitory concentrations of antibiotics. However, is there evidence that the concentrations cited in this paragraph are high enough to cause any effects? It would be helpful if the two paragraphs could be connected in this way. Is 21.4 ng/L really high enough to have any biological effect? We are not told.

Line 298-299: The author refers to antibiotic resistant bacteria being transferred from food animals to humans and cite reference 108 Kummerer. This is a rather controversial area, and while there is plenty to fear for such transfer, the article cited is another review article discussing the potential. If there is direct evidence of transfer of ARB from animals to humans, the author should say so and cite it. If the evidence is not there, the author should refer to the potential for transfer and not imply such transfer actually occurs without backing it up.

Round 2

Reviewer 2 Report

The paper has now improved with some revisions by the authors. Very few changes in style and formatting still required, to be solved in copy editing phase.

The paper can be accepted